# Association between self-reported and proxy informant Health Quality of life of older adults with the use of antipsychotic drugs in nursing homes. A cross-sectional study

**Denis Boucaud-Maitre**[1,2]*, **Fares Jaballah**[1], **Luc Letenneur**[3], **Leila Rinaldo**[4], **Jean-François Dartigues**[3], **Moustapha Dramé**[2,5], **Hélène Amieva**[3], **Maturin Tabué-Teguo**[2,5]

**1** DRCI, Centre Hospitalier le Vinatier, Bron, France, **2** Equipe EPICLIV, Université des Antilles, Fort-de-France, Martinique, **3** Inserm U1219 Bordeaux Population Health Center, University of Bordeaux, Bordeaux, France, **4** Centre Hospitalo-Universitaire de Guadeloupe, Pointe-à-Pitre, Guadeloupe, **5** Centre Hospitalo-Universitaire de Martinique, Fort-de-France, Martinique

* denis.boucaud@gmail.com

## Abstract

### Introduction

Antipsychotic prescriptions are frequent in nursing homes due to the challenging management of symptoms associated with Alzheimer's disease and related neurodegenerative disorders. This study aimed to assess the association between Health-related Quality Of Life (HrQOL) and antipsychotic use in nursing homes.

### Methods

This is a cross-sectional study of the KASEHPAD (Karukera Study of Ageing in Nursing Homes) study conducted in six nursing homes in Guadeloupe and Martinique (French West Indies). The EuroQol 5-dimensional questionnaire (EQ-5D) was used to measure HRQoL. Clinical characteristics and use of antipsychotic drugs of 194 older adults with both self-reported and proxy informant HrQOL index were extracted. Other outcomes measures included the frequency and severity of delusions, hallucinations and agitation using the reduced Neuropsychiatric Inventory Nursing Home (NPI-R) score, Activities of Daily Living (ADL) score and Mini-Mental State Examination (MMSE) score.

### Results

The mean age of participants was 81.3 years and 63.6% had major cognitive impairment (MMSE score ≤18). The prevalence of delusions (39.7%), hallucinations (27.8%) and agitation (40.7%) was high. Antipsychotic medication was prescribed to 37.1% of the participants. The self-reported HrQOL Index of older adults was higher than the proxy HrQOL Index (0.54 ± 0.43 versus 0.40 ± 0.43) with a correlation coefficient of 0.63 (p<0.001). The use of antipsychotic medication was associated with a higher self-reported HrQOL index, after controlling for the frequency ($\beta$ = 0.144, p = 0.024) or severity ($\beta$ = 0.159, p = 0.009) of

**Data Availability Statement:** All relevant data are within the paper and its Supporting information files.

**Funding:** This study was supported by a grant from the Conseil Départemental de la Guadeloupe and ARS de la Guadeloupe, Saint-Martin, and Saint-Barthélemy (grant 2020/DPAPH/DRM) and ARS Martinique. The funding body had no role in the design of the study and collection, analysis, and interpretation of data and in writing the manuscript.

**Competing interests:** The authors have declared that no competing interests exist.

delusions, hallucinations and agitation symptoms. Conversely, antipsychotic use was not associated with proxy HrQOL Index.

## Conclusion

Despite the adverse effects of long-term use of antipsychotic drugs in older adults, it is associated with better-perceived HrQOL among nursing home residents. However, this association was not observed when HrQOL was assessed by proxy informants. This finding may explain the challenges in reducing the use of this therapeutic class in nursing homes.

## Introduction

Despite numerous warnings from national and international guidelines, the extensive use of antipsychotic drugs for managing hallucinations, delusions and agitation can make it difficult for carers to deal with Alzheimer's disease and similar neurodegenerative conditions in nursing homes [1]. In fact, the use of antipsychotics in Western European nursing homes range from 12% to 59% [2]. In the United States (US), the prevalence of antipsychotic use is 17.2% in patients with Alzheimer's disease and related dementias, compared to 6.6% in those without such conditions [3]. However, the use of antipsychotics is associated with an increased risk of adverse effects, such as stroke, myocardial infarction, and death [4]. According to a study by Nielsen R.E et al. [5], current exposure to antipsychotics is associated with increased mortality rates, with Hazard Ratios (HR) ranging from 1.92 to 2.31 for deaths from cardiovascular disease, cancer and infections. In a Japanese study of 10,079 older adults with Alzheimer's disease (women, 69%; mean age, 81 years) [6], first-time antipsychotic users had increased mortality, with 9.4% dying during the 11–24 week period compared to 2% in the not exposed to antipsychotic drugs group. The authors suggested that recent use of antipsychotics is an independent risk factor for mortality. Additionally, a literature review by Corbett. A et al. [7], based on 18 studies, highlighted an increased risk of cognitive decline after 12 weeks of antipsychotic treatment. This review also reported a 3-fold excess risk of cerebrovascular events, an increased frequency of extrapyramidal symptoms, peripheral edema, sedation, QTc prolongation, infections and gait disorders.

The risk-benefit balance of antipsychotic use in dementia remains controversial. Some evidence suggests that typical antipsychotics may slightly reduce agitation and psychosis in people with dementia [8]. However, due to the associated adverse events, several countries have restricted their use in patients with dementia. Clinical guidelines from the UK National Institute for Health and Care Excellence state that antipsychotics should only be used when other approaches have proved insufficient, and the patients are in severe distress or at risk of harming themselves or others [4]. The FDA recommendations state that "antipsychotics are not indicated for the treatment of dementia-related psychosis", except in cases of onset of delirium [9]. Nevertheless, antipsychotics continue to be widely used in nursing homes to manage the behavioral and psychological symptoms of dementia. In the US, there are discrepancies in the use of antipsychotics and qualifying diagnoses in nursing homes [9]. In 2023, the Biden administration announced efforts to curb the inappropriate use of antipsychotics in nursing homes based on unfounded diagnoses. As part of this initiative, the US Department of Health and Human Services should conduct targeted audits of nursing homes to determine whether they are misdiagnosing schizophrenia in their residents to justify the use of antipsychotic drugs [10].

Agitation and psychotic symptoms are distressing for older adults with dementia and their caregivers. Such psychotic symptoms are also associated with impaired quality of life (QOL) in nursing homes [11]. However, the benefit of antipsychotic on residents' QOL remains uncertain. Some studies have reported that antipsychotic use in nursing homes is associated with a lower QOL, as assessed by tools such as the QUALIDEM [12] or Dementia Care Mapping [13]. WHO defines quality of life as "individuals' perceptions of their position in life in the context of the culture and value systems in which they live and in relation to their goals, expectations, standards and concerns" [14]. QOL is multimodal but the subjective measurement of health-related quality of life (HrQOL) has become a valuable dimension in assessing the benefits of health interventions from the patient's perspective. It is considered as an essential indicator of the overall health status [15]. According to a Cochrane systematic review, no randomized study on psychotic drug reported having measured HrQOL in people with Alzheimer's disease and vascular dementia [8]. In a study on 272 community-dwelling and 123 institutionalized patients [16], Hessmann *et al.* observed that use of antipsychotics was not associated with self-reported or proxy HrQOL after adjustments for age, gender, living situation, MMSE and depression. However, this study did not adjust for the presence and severity of psychotic symptoms.

Given the high rate of antipsychotic medication in nursing homes, it is crucial to understand the reasons for this overprescribing. There is probably a disparity in perception between clinicians working in nursing homes who regularly prescribe antipsychotics, patients' views on the usefulness of their treatment, and health authorities advocating for substantial restrictions on this class of medication. Nevertheless, the shared objective of all parties is to maintain or improve the quality of life of the residents. The study of the association between quality of life and antipsychotic use could facilitate the development of shared decision-making processes, in which patients and clinicians collaborate to make treatment choices based on the patients' preferences and the best available evidence [17].

We hypothesized that antipsychotic medication could impact HrQOL, considering the frequency and severity of psychotic symptoms. Additionally, we hypothesized that the assessment of HrQOL reported by residents and their respective carers (proxy) might differ. Indeed, residents with varying levels of pain, depression or cognitive impairment, tend to provide discordant responses to HrQOL [18]. This study aimed to assess the association between self-reported and proxy HrQOL and the use of antipsychotics in Caribbean nursing homes.

## Methods

In this study, we analyzed baseline data collected as part of the KASEHPAD (KArukera Study of Ageing in EHPAD) study [19], a one-year prospective cohort conducted in 6 nursing homes in Guadeloupe and Martinique (French West Indies). Participants were eligible for inclusion if they were aged 60 years or older, resided in nursing homes and were covered by French social security. A total of 332 residents were enrolled between September 2020 and November 2022. Healthcare professionals, including experienced geriatricians and clinical research nurses, conducted interviews with the participants and their professional caregivers at baseline. The participant interview included a physical examination (assessing grip strength, gait speed, and blood pressure), a cognitive evaluation (Mini Mental State Exam test), and a psychological assessment incorporating the EQ-5D-3L scale. The estimated completion time was 30 minutes but varied according to the participant's physical and cognitive condition. Professional caregivers provided detailed information on various aspects of the participants, including sociodemographic data, medical history, medication, nutritional status, level of

activity, degree of dependence (The Katz Activities of Daily Living), and a proxy EQ-5D-3L scale. For the purpose of this study, we extracted only participants for whom both self-reported and informant-rated (proxy) EuroQol 5-Dimensional questionnaire (EQ-5D) were available (n = 194, 58.4% of participants).

## Ethics

This study was registered on clinicaltrials.gov (NCT04587466) and received approval from the EST-1 French Ethics Committee (ID RCB: 2020-A00960-39). The KASEHPAD study is an observational study involving human participants, with no identified risks to participant safety. Consequently, in accordance with the regulatory framework (Law 2012–300), the requirement for a signed consent form has been waived by the relevant authorities. Residents have received an informational leaflet outlining the key aspects of the study and have provided oral consent. Participants received an informational leaflet outlining the key aspects of the study. Participation was entirely voluntary, and individuals had the option to decline participation or withdraw at any point without facing any negative consequences. Considering the high prevalence of cognitive impairment among nursing home residents, the on-site investigator took care to ensure that participants fully comprehended the implications of their involvement. In instances where a participant was under legal guardianship and/or unable to understand the study, non-opposition was obtained from the legal guardian or a designated contact person. All data were anonymized.

## Measures

**Health related quality of life..**   HRQoL was assessed using the EQ-5D-3L questionnaire. The EQ-5D scale is a globally recognized instrument for assessing quality of life and has been extensively utilized across various countries [20]. The EQ-5D-3L questionnaire is also considered a valuable predictor of mortality and initial hospitalization in older populations [21]. It comprises five dimensions (mobility, self-care, usual activities, pain/discomfort, and anxiety/depression) and three response levels ('no problems', 'some problems', and 'extreme problems'). Both the older adults (self-reported EQ-5D) and paramedical informants (proxy EQ-5D) completed the scale. An individual health profile (self-reported HrQOL index and proxy HrQOL index) was assigned using a summary index score based on societal preference weights for health status [22]. The French value set for the EQ-5D-3L was used to calculate the EQ-5D Index [19], with values below zero indicating a health state worse than death and 1 indicating full health.

**Psychotic symptoms.**   Delusions, hallucinations and agitation were measured using the Reduced Neuropsychiatric Inventory (NPI-R) [23]. The Neuropsychiatric Inventory (NPI) [24] is one of the most widely utilized scales for assessing symptoms in individuals with dementia and other neurological disorders. Originally developed for populations with dementia, the NPI has also been employed in asessing patients with psychotic, affective, and other neurological conditions, including Parkinson's disease [25]. This scale assesses 12 behavioral domains: delusions, hallucinations, agitation and aggression, depression and dysphoria, anxiety, euphoria and elation, apathy and indifference, disinhibition, irritability and lability, aberrant motor behavior, nighttime behavioral disturbances, and appetite and eating changes. For each domain, the informant was asked to provide corresponding ratings for frequency (present or absent) and severity (3-point Likert-type scale) of symptoms experienced within the past week. The results of the NPI-R correlate with those of the longer version, but its administration time is shortened [23].

## Antipsychotic drug

Antipsychotic drugs were classified under the group N05A in the Anatomical Therapeutic Chemical (ATC) classification system. These drugs were further categorized into two sub-groups: atypical antipsychotics (including amisulpride, aripiprazole, clozapine, olanzapine, paliperidone, quetiapine, and risperidone) and other antipsychotics (comprising cyamemazine, zuclopenthixol, pipamperone, haloperidol, propericiazine, levomepromazine, pipotiazine, penfluridol, and sulpiride).

**Other measures.** The following variables were also extracted: age, gender and comorbidities (presence or absence of diabetes, hypertension, dementia, depression, Parkinson disease, chronic pain). Cognitive performance was assessed using the Mini Mental State Exam (MMSE) tool [26] generating a score from 0 to 30, where a low score indicates poor cognitive function. The Katz Activities of Daily Living (ADL) was used to assess functional status [27].

## Statistical analysis

Quantitative variables were expressed as mean ± standard deviation (SD). Qualitative variables were expressed as percentages. A t-test or Spearman correlation test was used to compare the characteristics of the residents and their self-reported or proxy EQ-5D Indexes. Multiple linear regression was used to estimate the association between HrQoL Index (self-reported or proxy) and antipsychotic use, age, sex, chronic pain, ADL score, MMSE score, and controlling either the frequency (model 1) or the severity (model 2) of delusion, hallucination and agitation. Severity was considered as a binary categorical variable (no or mild disorder versus moderate to severe disorder). Missing values were not imputed. All analyses were performed using R v.4.0.2 software.

## Results

A total of 194 older adults living in Caribbean nursing homes who completed the EQ5D scale, either though self-report or proxy, were included in the study. The mean age was 81.3 years with half being male. The prevalence of diabetes and hypertension was 29.9% and 69.6%, respectively. Among the older adults, 46.4% had dementia and 63.6% had a MMSE score ≤18 indicating major cognitive disorders. Additionally, 9.8% had Parkinson's disease. Delusions (39.7%), hallucinations (27.8%) and agitation (40.7%) were common symptoms. Psychotropic medication was prescribed to 37.1% of participants, half of whom were taking atypical antipsychotics (Table 1). The self-reported HrQOL Index was higher than the proxy HrQOL Index (0.54 ± 0.43 versus 0.40 ± 0.43) and the correlation test was 0.63 (p<0.001).

### Association of antipsychotic use with self-reported HrQoL

We first investigated whether HrQOL was associated with antipsychotic medication from the perspective of older adults. The bivariate analysis revealed a statistically significant association between higher HrQOL Index scores and the following factors: younger age (r = -0.23; p = 0.001), lower levels of dependency (r = 0.54; p < 0.001), better cognitive function (r = 0.146; p = 0.046), male gender (0.61 ± 0.38 vs. 0.48 ± 0.46; p = 0.035), absence of chronic pain (no pain: 0.61 ± 0.41 vs. pain: 0.46 ± 0.43; p = 0.011), and antipsychotic medication use (0.67 ± 0.36 vs. no use: 0.46 ± 0.45; p < 0.001). In the model adjusted for the frequency (model 1) of delusion, hallucination and agitation symptoms, antipsychotic use (β = 0.144, p = 0.024, p<0.001) and ADL score (β = 0.126, p<0.001) were significantly associated with a higher HrQOL index. Conversely, the MMSE score was marginally negatively associated with HrQOL index (β = -0.009, p = 0.045) (Table 2).

**Table 1. Characteristics of the older adults living in Caribbean nursing homes.** N = 194.

| Characteristics | N(%) or Mean (SD); n |
|---|---|
| Age | 81.3 (10.2); 194 |
| Sex (Male) | 93 (47.9%); 194 |
| Diabetes | 58 (29.9%); 194 |
| Hypertension | 135 (69.6%); 194 |
| Dementia | 90 (46.4%); 194 |
| Parkinson's disease | 19 (9.8%); 194 |
| Depression | 42 (21.6%); 194 |
| Chronic pain | 89 (45.9%); 194 |
| ADL score | 3.2 (2.0); 194 |
| MMSE score | 16.0 (7.4); 187 |
| MMSE score ≤18 | 119 (63.6%); 187 |
| Delusion symptoms | 77 (39.7%); 194 |
| Hallucination symptoms | 54 (27.8%); 194 |
| Agitation symptoms | 79 (40.7%); 194 |
| Antipsychotic use | 72 (37.1%); 194 |
| • Atypical antipsychotic | 37 (19.1%); 194 |
| • Other antipsychotic | 35 (18.0%); 194 |
| Antipsychotic use by MMSE score | |
| MMSE score > 18 | 44.1%; 68 |
| MMSE score range 11–18 | 31.1%; 74 |
| MMSE score ≤ 10 | 33.3%; 45 |
| Self-reported HrQOL Index | 0.54 (0.43); 194 |
| Proxy-reported HrQOL Index | 0.40 (0.39); 194 |

The results were similar when the severity (model 2) of delusion, hallucination and agitation symptoms was considered. HrQOL Index was associated with antipsychotic use (β = 0.159, p = 0.009), ADL score (β = 0.127, p<0.001) and MMSE score (β = -0.010, p = 0.024).

## Association of antipsychotic use with proxy HrQoL

We further investigated whether similar associations were present when using informant-rated HrQOL. In bivariate analysis, a higher proxy HrQOL Index was associated with younger age (r = -0.367; p = 0.001), a lower level of dependency (r = 0.82; p<0.001), male gender (0.51 ± 0.37 versus 0.30 ± 0.38; p<0.001), better cognitive function (r = 0.395; p<0.001), antipsychotic use (0.50 ± 0.38 versus no use: 0.33 ± 0.39; p = 0.004), absence of chronic pain (0.48 ± 0.38 versus pain: 0.29 ± 0.38; p<0.001) and absence of hallucination symptoms (0.44 ± 0.37 versus 0.28 ± 0.42; p = 0.016). In the adjusted models that included either the frequency (model 1) or severity (model 2) of delusion, hallucination and agitation symptoms, only ADL score remained significantly associated with the HrQOL Index (β = 0.154, p<0.001). Antipsychotic use was not associated with the HrQOL Index ((β = 0.041, p = 0.278) (Table 3).

## Discussion

Our study suggests that from the perspective of older adults living in nursing home, the use of antipsychotics was associated with better HrQOL. This result remained unchanged whether bivariate analysis or multivariate analysis was used, considering the frequency or severity of

**Table 2. Bivariate and adjusted linear regression on the relationship of self-reported HrQOL index with antipsychotic use.**

| | n | Bivariate analysis | | Model 1 (controlled for the frequency of psychotic symptoms) | | Model 2 (controlled for the severity of psychotic symptoms) | |
|---|---|---|---|---|---|---|---|
| | | Correlation coefficient or mean ± SD | p | Estimate (CI95%) | p | Estimate (CI95%) | p |
| Age | 194 | -0.23 | 0.001 | - | 0.705 | - | 0.766 |
| Sex | | | | | | | |
| Male | 93 | 0.61 ± 0.38 | 0.035 | - | 0.722 | - | 0.671 |
| Female | 101 | 0.48 ± 0.46 | | | | | |
| ADL score | 194 | 0.54 | <0.001 | 0.126 (0.094, 0.158) | <0.001 | 0.127 (0.095,0.158) | <0.001 |
| MMSE score | 187 | 0.146 | 0.046 | -0.009 (-0.017–0.000) | 0.040 | -0.010 (-0.018, -0.001) | 0.024 |
| Antipsychotic drugs use | | | | | | | |
| Yes | 75 | 0.67 ± 0.36 | <0.001 | 0.144 (0.027, 0.261) | 0.016 | 0.159 (0.042, 0.276) | 0.009 |
| No | 119 | 0.46 ± 0.45 | | | | | |
| Chronic Pain | | | | | | | |
| Present | 89 | 0.46 ± 0.43 | 0.011 | - | 0.448 | - | 0.480 |
| Absent | 105 | 0.61 ± 0.41 | | | | | |
| Delusion (Frequency) | | | 0.860 | - | 0.862 | | |
| Present | 77 | 0.55 ± 0.45 | | | | | |
| Absent | 117 | 0.54 ± 0.41 | | | | | |
| Delusion (Severity) | | | | | | - | 0.409 |
| 0–1 | 154 | 0.52 ± 0.43 | 0.223 | | | | |
| 2–3 | 40 | 0.62 ± 0.44 | | | | | |
| Hallucination | | | | - | 0.382 | | |
| Present | 54 | 0.49 ± 0.25 | 0.348 | | | | |
| Absent | 140 | 0.56 ± 0.16 | | | | | |
| Hallucination (Severity) | | | | | | - | 0.251 |
| 0–1 | 166 | 0.56 ± 0.42 | 0.346 | | | | |
| 2–3 | 28 | 0.46 ± 0.25 | | | | | |
| Agitation | | | | - | 0.211 | | |
| Present | 79 | 0.61 ± 0.43 | 0.082 | | | | |
| Absent | 115 | 0.50 ± 0.43 | | | | | |
| Agitation (Severity) | | | | | | - | 0.917 |
| 0–1 | 160 | 0.54 ± 0.43 | 0.860 | | | | |
| 2–3 | 34 | 0.55 ± 0.45 | | | | | |

psychotic symptoms. This finding may explain the challenges in reducing the use of this therapeutic class in nursing homes. Antipsychotic drugs continues to be widely used due to their perceived efficacy in managing agitation and hallucinations, albeit potentially limited, and their relatively good tolerance in the short-term. Their impact on mortality and other adverse events may be challenging to discern for older adults primarily affected by dementia and multiple comorbidities. However, antipsychotic treatment is often prolonged chronically despite the lack of indication for its continuation in many patients.

We also observed that the use of antipsychotic medication was not associated with better HrQOL from staff's perspective. Training staff could play an important role in reducing the use of antipsychotic medication if they understand their potential harms and are involved in the appropriate management of psychotic symptoms. Furthermore, professional caregivers are in contact with patients and can serve as a link with prescribing physicians who do not see the patients on a daily basis. HrQOL was associated with hallucinations in bivariate analysis for

**Table 3. Bivariate and adjusted linear regression on the relationship of proxy HrQOL Index with antipsychotic use.**

| | n | Bivariate analysis | | Model 1 (controlled for the frequency of psychotic symptoms) | | Model 2 (controlled for the severity of psychotic symptoms) | |
|---|---|---|---|---|---|---|---|
| | | Correlation coefficient or mean ± SD | p | Estimate | p | Estimate | p |
| Age | 194 | -0.367 | 0.001 | - | 0.847 | - | 0.937 |
| Sex | | | | | | | |
| Male | 93 | 0.51 ± 0.37 | <0.001 | - | 0.104 | - | 0.095 |
| Female | 101 | 0.30 ± 0.38 | | | | | |
| ADL | 194 | 0.82 | <0.001 | 0.154 (0.134, 0.175) | <0.001 | 0.154 (0.134, 0.175) | <0.001 |
| MMSE | 187 | 0.395 | <0.001 | - | 0.943 | - | 0.887 |
| Antipsychotic drugs use | | | | 0.038 | 0.313 | 0.041 | 0.278 |
| Yes | 75 | 0.50 ± 0.38 | 0.004 | | | | |
| No | 119 | 0.33 ± 0.39 | | | | | |
| Chronic Pain | | | | - | 0.106 | | |
| Present | 89 | 0.29 ± 0.38 | <0.001 | | | - | 0.122 |
| Absent | 105 | 0.48 ± 0.38 | | | | | |
| Delusion (Frequency) | | | 0.536 | - | 0.320 | | |
| Present | 77 | 0.38 ± 0.40 | | | | | |
| Absent | 117 | 0.41 ± 0.39 | | | | | |
| Delusion (Severity) | | | | | | - | 0.398 |
| 0–1 | 154 | 0.41 ± 0.39 | 0.223 | | | | |
| 2–3 | 40 | 0.39 ± 0.36 | | | | | |
| Hallucination | | | | - | 0.105 | | |
| Present | 54 | 0.28 ± 0.42 | 0.016 | | | | |
| Absent | 140 | 0.44 ± 0.37 | | | | | |
| Hallucination (Severity) | | | | | | - | 0.340 |
| 0–1 | 166 | 0.42 ± 0.38 | 0.027 | | | | |
| 2–3 | 28 | 0.24 ± 0.40 | | | | | |
| Agitation | | | | - | 0.971 | | |
| Present | 79 | 0.40 ± 0.35 | 0.904 | | | | |
| Absent | 115 | 0.39 ± 0.42 | | | | | |
| Agitation (Severity) | | | | | | - | 0.419 |
| 0–1 | 160 | 0.41 ± 0.40 | 0.291 | | | | |
| 2–3 | 34 | 0.33 ± 0.37 | | | | | |

caregivers, but not for residents. This finding highlights the difficulty that caregivers have in dealing with hallucinations, although older adults appear to be less affected. However, in the multivariate analysis, we found that only dependency had an impact on older adults HrQOL from the caregivers' perspective, whereas antipsychotic use, dependency and cognition were associated with HrQOL according to the residents.

Our finding should be seen in the context of the study by Hessmann *et al.* [16]. In this retrospective cohort study involving patients with dementia, aged ≥65 years, the authors suggest limited efficacy of antipsychotics for managing aggression and psychosis. A similar trend was observed for HrQOL, as in multivariate analysis, there was no significant effect of antipsychotics on proxy HrQOL (OR: 0.46 [CI95%: 0.13–1.55], p = 0.210) and a favorable trend for self-reported HrQOL (OR: 3.58 [CI95%: 0.94–13.67]). In our study, we adjusted for the frequency or the severity of psychotic symptoms to align more closely with the recommendations for antipsychotic use.

QOL is multidimensional and health perceptions represent only one aspect of it. Autonomy, role and activity, relationships, spirituality, emotional comfort or adaptation are also crucial components [28]. The EQ5D scale focuses on health perceptions and is not correlated with life satisfaction for example [29]. Various QOL scales have been used in other studies, and findings regarding the association between antipsychotic use and QOL are inconsistent. Some studies have reported associations between antipsychotic use and lower QOL [12, 13], while others have found no such an association [11]. Our results are therefore relevant as they suggest that older adults may perceive a subjective clinical benefit of antipsychotic medication without necessarily experiencing a benefit to their global quality of life. This underscores the complexity of assessing QOL in individuals receiving antipsychotic treatment and highlights the need for further research to comprehensively understand its impact.

Our study has several strengths. We used both self-reported and proxy EQ-5D scales, revealing disparities in perceptions of HrQOL were not similar with the use of antipsychotic drug, despite a good correlation between the two. Proxy EQ5D is frequently used in clinical studies in nursing homes because due to the high prevalence of dementia among residents, who are unable to complete the scale themselves. While nursing staff assessment of HRQoL provides a more complete dataset, there remains considerable uncertainty regarding whether these scores adequately represent the diverse groups of residents in nursing homes [18]. In our study, we adjusted for the frequency or severity of hallucinations, delusions and agitation with similar results. Additionally, we adjusted for ADL scores, as the level of dependency has been strongly associated with QOL [30, 31]. However, our study had also limits. We only included older adults with self-reported EQ-5D, therefore excluding those with severe dementia who were unable to complete the scale. This aspect warrants consideration, as our findings may not fully represent the entire nursing home population. Furthermore, our study needs to be confirmed with longitudinal cohorts, particularly as potential sociocultural factors may be specific to the Caribbean population. Indeed, Caribbean populations remain understudied, especially in the fields of gerontology and psychogeriatrics. Since it is not unequivocal which symptoms may serve as potential confounders, the regression analyses were not adjusted for the influence of other neuropsychiatric symptoms, notably sleep disturbance. At last, this study is a cross-sectional study and we lack information on the duration and adherence to antipsychotic prescriptions. Consequently, it is challenging to draw conclusions about the positive or negative effects of antipsychotics independently of the prescription duration.

## Conclusion

The present findings underscore the association between antipsychotics and HrQOL contingent upon the perceptions of the older adult or their caregiver. A higher HrQOL index was associated with the use of antipsychotic medication, considering the severity or frequency of delusions, hallucinations and agitation. Our study contributes to the existing literature by suggesting that antipsychotic drugs play a role in older adult's perception of HrQOL. There is a need to involve the older adults in the management of their psychotic symptoms and to explain the potential harm of antipsychotics. Further research in other populations is warranted to ascertain whether this interesting finding is a widespread phenomenon.

## Supporting information

**S1 Dataset.**
(CSV)

**S1 Checklist. Human participants research checklist.**
(DOCX)

## Acknowledgments

We would like to thank the ACTIVE Team from Bordeaux for their precious methodological support, as well as Valérie Soter, and Mélanie Petapermal for their regulatory support. We thank the following nursing homes for participating in the study: EHPAD Les Flamboyants (Gourbeyre, Guadeloupe), EHPAD Kalana (Bouillante, Guadeloupe), EHPAD Nou Grand Moun (Capesterre-Belle-Eau, Guadeloupe), EHPAD les Jardins de Belost (Saint-Claude, Guadeloupe), Centre Hospitalier Gerontologique Palais Royal (Les Abymes, Guadeloupe) and Centre Emma Ventura (Fort-de-France, Martinique).

## Author Contributions

**Conceptualization:** Denis Boucaud-Maitre, Jean-François Dartigues, Hélène Amieva, Maturin Tabué-Teguo.

**Formal analysis:** Denis Boucaud-Maitre.

**Funding acquisition:** Moustapha Dramé, Maturin Tabué-Teguo.

**Investigation:** Denis Boucaud-Maitre, Leila Rinaldo, Maturin Tabué-Teguo.

**Methodology:** Denis Boucaud-Maitre, Jean-François Dartigues, Hélène Amieva, Maturin Tabué-Teguo.

**Project administration:** Denis Boucaud-Maitre.

**Resources:** Hélène Amieva.

**Supervision:** Jean-François Dartigues, Moustapha Dramé, Maturin Tabué-Teguo.

**Validation:** Jean-François Dartigues, Moustapha Dramé, Hélène Amieva, Maturin Tabué-Teguo.

**Writing – original draft:** Denis Boucaud-Maitre, Fares Jaballah.

**Writing – review & editing:** Luc Letenneur, Leila Rinaldo, Jean-François Dartigues, Moustapha Dramé, Hélène Amieva, Maturin Tabué-Teguo.

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
