## [Decision Letter · Decision Letter 0]

1 Aug 2024

PONE-D-24-26842Association between self-reported and proxy informant Health Quality of life of older adults with the use of antipsychotic drugs in nursing homes. A cross-sectional studyPLOS ONE

Dear Dr. Boucaud-Maitre,

Thank you for submitting your manuscript to PLOS ONE. After careful consideration, we feel that it has merit but does not fully meet PLOS ONE’s publication criteria as it currently stands. Therefore, we invite you to submit a revised version of the manuscript that addresses the points raised during the review process.

**Please address the comments from the reviewers as under prior to further consideration for publication**

We look forward to receiving your revised manuscript.

Kind regards,

Souparno Mitra, M.D.

Academic Editor

PLOS ONE

Journal Requirements:

This study was supported by a grant from the Conseil Départemental de la Guadeloupe and ARS de la Guadeloupe, Saint-Martin, and Saint-Barthélemy (grant 2020/DPAPH/DRM) and ARS Martinique. The funding body had no role in the design of the study and collection, analysis, and interpretation of data and in writing the manuscript.

3. In the online submission form, you indicated that The raw data supporting the conclusions of this article will be made available by the authors, without undue reservation.

Reviewers' comments:

Reviewer's Responses to Questions

**Comments to the Author**

1. Is the manuscript technically sound, and do the data support the conclusions?

Reviewer #1: Yes

Reviewer #2: Yes

2. Has the statistical analysis been performed appropriately and rigorously? 

Reviewer #1: Yes

Reviewer #2: Yes

3. Have the authors made all data underlying the findings in their manuscript fully available?

Reviewer #1: Yes

Reviewer #2: Yes

4. Is the manuscript presented in an intelligible fashion and written in standard English?

Reviewer #1: Yes

Reviewer #2: Yes

5. Review Comments to the Author

**Reviewer #1: **Your study highlights the complexity of managing antipsychotic use in nursing homes, emphasizing the need to consider diverse perspectives to improve patient care. Recognizing the multifaceted nature of quality of life and using varied assessment tools can provide a more comprehensive understanding of antipsychotic medication effects. The focus on this complexity is essential for advancing the field and enhancing patient care. Additionally, your study offers valuable insights into the impact of antipsychotic medications on nursing home residents' quality of life, highlighting both strengths and limitations. Implementing recommendations for future research can contribute to improved care for individuals in nursing homes.

**Reviewer #2: **Thank you for the opportunity to review the manuscript. The manuscript is well written, the methodology is sound, and the findings are clearly articulated, providing valuable insights into the potential use of antipsychotic drugs on the quality of life of older adults in nursing homes. However, there are a few areas that need revisions for clarity and completeness.

1. Introduction : Emphasize the importance of the study’s findings for clinical practice and policy-making in nursing homes, explaining how the results can influence the management of antipsychotic use.

2. Methods : Provide a rationale for the selection of EQ-5D-3L and NPI-R measurement tools, explaining why they are appropriate for assessing HRQoL and psychotic symptoms in this study population.

3. Describe the data collection process in detail. Add details about the timing (e.g., frequency of assessments, duration of each session) and methods (structured interviews, use of standardized questionnaires) used to collect data from participants and proxies.

4. Expand the ethics section to include details on how data confidentiality was maintained and any measures taken to ensure the welfare of participants during the study.

5.Results : Clearly indicate the statistical significance of the results within the text by mentioning p-values for all relevant comparisons and analyses. This will help readers quickly understand which results are statistically significant without having to refer to the tables.

6. Identify and discuss any potential confounding variables (such as severity of cognitive impairment, presence of comorbid conditions, medication adherence, and nursing home environment) that could have impacted the study’s outcomes.

6. PLOS authors have the option to publish the peer review history of their article (what does this mean?). If published, this will include your full peer review and any attached files.

Reviewer #1: No

Reviewer #2: **Yes: **Mohsin Raza

---

## [Author Response · Author response to Decision Letter 0]

23 Aug 2024

Reviewer #1: Your study highlights the complexity of managing antipsychotic use in nursing homes, emphasizing the need to consider diverse perspectives to improve patient care. Recognizing the multifaceted nature of quality of life and using varied assessment tools can provide a more comprehensive understanding of antipsychotic medication effects. The focus on this complexity is essential for advancing the field and enhancing patient care. Additionally, your study offers valuable insights into the impact of antipsychotic medications on nursing home residents' quality of life, highlighting both strengths and limitations. Implementing recommendations for future research can contribute to improved care for individuals in nursing homes.

Authors comment: We thank the reviewer for her/his encouraging comments and its interest on our work.

Reviewer #2: Thank you for the opportunity to review the manuscript. The manuscript is well written, the methodology is sound, and the findings are clearly articulated, providing valuable insights into the potential use of antipsychotic drugs on the quality of life of older adults in nursing homes. However, there are a few areas that need revisions for clarity and completeness.

1. Introduction : Emphasize the importance of the study’s findings for clinical practice and policy-making in nursing homes, explaining how the results can influence the management of antipsychotic use.

Authors comment: First and foremost, we would like to thank the reviewer for their comments, all of which we have taken into consideration. We agree with the reviewer’s comment 

1. We have added this point in the introduction section:

“There is a disparity in perception between clinicians working in nursing homes who regularly prescribe antipsychotics, patients’ views on the usefulness of their treatment, and health authorities advocating for substantial restrictions on this class of medication. Nevertheless, the shared objective of all parties is to maintain or improve the quality of life of the residents. The study of the association between quality of life and antipsychotic use could facilitate the development of shared decision-making processes, in which patients and clinicians collaborate to make treatment choices based on the patients’ preferences and the best available evidence.”

With the following reference:

Haugom EW, Benth JŠ, Stensrud B, Ruud T, Clausen T, Landheim AS. Shared decision making and associated factors among patients with psychotic disorders: a cross-sectional study. BMC Psychiatry. 2023;23(1):747. Published 2023 Oct 13. doi:10.1186/s12888-023-05257-y

2. Methods : Provide a rationale for the selection of EQ-5D-3L and NPI-R measurement tools, explaining why they are appropriate for assessing HRQoL and psychotic symptoms in this study population.

Authors comment: We agree with the reviewer’s comment. We have to justify the use of these two tools in the method section. Considering our target population (older adults in nursing homes, with 74% suffering from major cognitive disorders), we consider that EQ5D-3L and NPI are both valid and relevant for our study. We have added this in the method section:

“The EQ-5D scale is a globally recognized instrument for assessing quality of life and has been extensively utilized across various countries. The EQ-5D-3L questionnaire is also considered a valuable predictor of mortality and initial hospitalization in older populations.” with the following reference:

Cavrini G, Broccoli S, Puccini A, et al. EQ-5D as a predictor of mortality and hospitalization in elderly people. Qual Life Res. 2012;21:269–80.

And for the NPI-R:

“The Neuropsychiatric Inventory (NPI) is one of the most widely utilized scales for assessing symptoms in individuals with dementia and other neurological disorders. Originally developed for populations with dementia, the NPI has also been employed in evaluating patients with psychotic, affective, and other neurological conditions, including Parkinson’s disease.” with the reference: 

Lai CK (2014) The merits and problems of Neuropsychiatric Inventory as an assessment tool in people with dementia and other neurological disorders. ClinIntervAging 9,1051–1061

3. Describe the data collection process in detail. Add details about the timing (e.g., frequency of assessments, duration of each session) and methods (structured interviews, use of standardized questionnaires) used to collect data from participants and proxies.

Authors comment: We agree with the reviewer’s comment. We have added in the method section: 

“Healthcare professionals, including experienced geriatricians and clinical research nurses, conducted interviews with the participants and their professional caregivers. The participant interview included a physical examination (assessing grip strength, gait speed, and blood pressure), a cognitive evaluation (MMSE test), and a psychological assessment incorporating the EQ-5D-3L scale. Professional caregivers provided detailed information on various aspects of the participants, including sociodemographic data, medical history, medication, nutritional status, level of activity, degree of dependence (ADL), and the EQ-5D-3L scale. The estimated completion time was 30 minutes but varied according to the participant’s physical and cognitive condition.”

4. Expand the ethics section to include details on how data confidentiality was maintained and any measures taken to ensure the welfare of participants during the study.

Authors comment: We agree with the reviewer’s comment. We have added this point in the ethics section:

“Participants received an informational leaflet outlining the key aspects of the study. Participation was entirely voluntary, and individuals had the option to decline participation or withdraw at any point without facing any negative consequences. Considering the high prevalence of cognitive impairment among nursing home residents, the on-site investigator took care to ensure that participants fully comprehended the implications of their involvement. In instances where a participant was under legal guardianship and/or unable to understand the study, non-opposition was obtained from the legal guardian or a designated contact person. All data were anonymized.”

5. Results : Clearly indicate the statistical significance of the results within the text by mentioning p-values for all relevant comparisons and analyses. This will help readers quickly understand which results are statistically significant without having to refer to the tables.

Authors comment: We agree with the reviewer’s comment. We have incorporated the statistical results into the text. Indeed, both the text and tables should be able to stand alone, allowing the reader to understand them independently.

6. Identify and discuss any potential confounding variables (such as severity of cognitive impairment, presence of comorbid conditions, medication adherence, and nursing home environment) that could have impacted the study’s outcomes.

Authors comment: We agree with the reviewer’s comment and added several limits to our study, notably medication adherence, and the impact of other NPS on HrQOL. The severity of cognitive impairment was a limitation we had previously identified.

---

## [Decision Letter · Decision Letter 1]

25 Sep 2024

Association between self-reported and proxy informant Health Quality of life of older adults with the use of antipsychotic drugs in nursing homes. A cross-sectional study

PONE-D-24-26842R1

Dear Dr. Boucaud-Maitre,

We’re pleased to inform you that your manuscript has been judged scientifically suitable for publication and will be formally accepted for publication once it meets all outstanding technical requirements.

Kind regards,

Souparno Mitra, M.D.

Academic Editor

PLOS ONE

Additional Editor Comments (optional):

Reviewers' comments:

Reviewer's Responses to Questions

**Comments to the Author**

1. If the authors have adequately addressed your comments raised in a previous round of review and you feel that this manuscript is now acceptable for publication, you may indicate that here to bypass the “Comments to the Author” section, enter your conflict of interest statement in the “Confidential to Editor” section, and submit your "Accept" recommendation.

Reviewer #1: All comments have been addressed

Reviewer #3: All comments have been addressed

2. Is the manuscript technically sound, and do the data support the conclusions?

Reviewer #1: Yes

Reviewer #3: Yes

3. Has the statistical analysis been performed appropriately and rigorously? 

Reviewer #1: Yes

Reviewer #3: Yes

4. Have the authors made all data underlying the findings in their manuscript fully available?

Reviewer #1: Yes

Reviewer #3: Yes

5. Is the manuscript presented in an intelligible fashion and written in standard English?

Reviewer #1: Yes

Reviewer #3: Yes

6. Review Comments to the Author

Reviewer #1: This article seeks to elucidate the significance of comprehending the intricate issue surrounding the utilization of antipsychotics in nursing homes for elderly individuals. It addresses the challenges inherent in evaluating the overall impact of these medications, particularly concerning mortality and adverse events in a population frequently afflicted with dementia and multiple comorbidities. The contrasting trends between proxy and self-reported Health-related Quality of Life (HrQOL) imply divergent perceptions of the effects of antipsychotic treatment by caregivers and residents. The article underscores the multifaceted nature of quality of life (QOL) and the constraints of relying solely on health perceptions to assess overall well-being. It emphasizes the indispensability of considering autonomy, relationships, and emotional comfort for a comprehensive understanding of QOL, particularly in the context of elderly individuals. Furthermore, it acknowledges inherent limitations in the study, such as sample bias, observational bias, lack of generalizability, and failure to account for other neuropsychiatric symptoms. The article advocates for addressing these limitations in future research to facilitate the development of effective interventions and enhance the quality of life for elderly individuals in nursing homes.

Reviewer #3: I think the manuscript does a good job of presenting solid research, and the data effectively backs up the conclusions. The statistical analysis seems well done, with appropriate controls and sample sizes that give confidence in the results. The writing is clear and straightforward, and I didn’t notice any major grammatical issues. I also appreciate that the authors have made all the underlying data fully available, which is in line with PLOS’s data policy. Overall, the study provides valuable insights into the link between antipsychotic use and quality of life in nursing homes, and I think it’s well-prepared for publication.

7. PLOS authors have the option to publish the peer review history of their article (what does this mean?). If published, this will include your full peer review and any attached files.

Reviewer #1: **Yes: **Rajasekhar Kannali

Reviewer #3: No

---

## [Editor Report · Acceptance letter]

25 Dec 2024

PONE-D-24-26842R1 

PLOS ONE

Dear Dr. Boucaud-Maitre, 

I'm pleased to inform you that your manuscript has been deemed suitable for publication in PLOS ONE. Congratulations! Your manuscript is now being handed over to our production team.

Kind regards, 

on behalf of

Dr. Souparno Mitra 

Academic Editor

PLOS ONE